# Breaking a Vicious Circle: Lymphangiogenesis as a New Therapeutic Target in Wound Healing

**DOI:** 10.3390/biomedicines11030656

**Published:** 2023-02-21

**Authors:** Filippo Renò, Maurizio Sabbatini

**Affiliations:** 1Health Sciences Department, Università del Piemonte Orientale, 28100 Novara, Italy; 2Sciences and Innovative Technology Department, Università del Piemonte Orientale, 15121 Alessandria, Italy

**Keywords:** lymphangiogenesis, wound healing, VEGF-A, VEGF-C/D, VEGFR-3

## Abstract

The lymphatic system is of fundamental importance in maintaining a fluid balance in the body and tissue homeostasis; it drains protein-rich lymph from the interstitial space and facilitates the release of cells that mediate the immune response. When one tissue is damaged, more cells and tissues work to repair the damaged site. Blood and lymph vessels are particularly important for tissue regeneration and healing. Angiogenesis is the process of the formation of new blood vessels and is induced by angiogenic factors such as VEGF-A; VEGF-C/D-induced lymphangiogenesis and both occur simultaneously during wound healing. After the inflammatory phase, lymphatic vessels suppress inflammation by aiding in the drainage of inflammatory mediators; thus, disorders of the lymphatic system often result in chronic and disabling conditions. It has recently been clarified that delayed wound healing, as in diabetes, can occur as a consequence of impaired lymphangiogenesis. In this review, we have highlighted recent advances in understanding the biology underlying lymphangiogenesis and its key role in wound healing, and the possibility of its pharmacological modulation as a novel therapeutic strategy for the treatment of chronic wounds.

## 1. Introduction

Lymphatic vascularization plays key roles in the tissue fluid balance, immune defense, general metabolism, and tumor metastases. The lymphatic system includes a series of lymphatic vessels through which a clear liquid called lymph flows, containing lipids, enzymes, hormones, different types of lymphocytes, and macrophages [1]. The lymph derives completely from the interstitial fluid, and it flows along the lymphatic vessels, reaching the lymph nodes, small bean-shaped masses of lymphoid tissue enclosed by a capsule of connective tissue. As part of the lymphatic system, they provide specialized tissues where foreign antigens present in the lymph can be trapped and exposed to cells of the immune system for destruction [2,3]. The lymphatic system returns to the venous circulation every day approximately 1–2 L of interstitial fluid (20–30 g of protein per liter) in a healthy adult subject [4], through blind lymphatic capillaries that drain liquids, macromolecules, and some leukocytes. Lymphatic capillary transport lymph to the larger collecting lymph vessels (the right lymphatic duct and the thoracic duct) to return the lymph to the bloodstream via the subclavian veins. Even partial interferences with lymphatic drainage can cause tissue swelling from excess fluid, a condition called edema.

Despite the importance of lymphatic drainage and immunological functions, not all organs are drained from the lymphatic system. In fact, the central nervous system, bones, bone marrow, maternal part of the placenta, endomysium of the muscles, crystalline, cornea, epidermis, cartilage, and inner tunic of large caliber arteries lack lymphatic vascularization [2]. Nevertheless, recent reports have confirmed the presence of meningeal lymphatic vessels (MLV) in the dura mater [2,5] and the existence of prelymphatic channels connected to the perivascular space able to drain effectively the interstitial fluid and assure the cleaning of the cerebral interstitial environment, the so-called glymphatic system [6]. Finally, even the retina [7,8,9], optic nerve [9,10,11], and conjunctive region [9,12,13] present MVL or glymphatic systems.

Interestingly, the exchange and removal of metabolites have shown great value and significance in the study of central nervous system diseases such as Alzheimer’s and subarachnoid hemorrhage, highlighting the importance of the lymphatic system in the central nervous system as well [9].

An additional function is performed in the small intestine, where the milky lymphatic vessels, present inside the intestinal villi, absorb the food lipids released by the intestinal epithelial cells in the form of chylomicrons.

Furthermore, the skin and mucous membranes are also particularly rich in lymphatic vessel tissues as they frequently come into contact with foreign antigens.

In adults, lymphatic vessel formation and remodeling occur primarily during inflammation, corpus luteum development, wound healing, and tumor growth.

Unlike the blood cardiovascular system, no pump provides the energy to circulate the lymph; therefore, in the lymphatic system, this flows slowly and at low pressure: the most important mechanism used to allow movement is given by the rhythmic contraction of the lymphatic vessels, and this happens when the fluid contained in them stretches the vessel wall. The propulsion of the lymph is also determined by the external compression of the lymphatic vessels given by the contraction of the skeletal muscles, as happens for venous circulation [14].

## 2. Embryology and Development of Lymphatic Vasculature

Lymphatic vessels begin to develop by the end of the fifth week of human embryonic development (Figure 1A). Distinct subpopulations of endothelial cells from the lateral parts of the anterior cardinal veins in the jugular region commit to the lymphatic lineage and then sprout laterally to form the first primordial lymphatic vascular structures, the primary lymph sac [15,16,17] (Figure 1B). Other lymph sacs will be then sprouted from major veins in different regions of the embryo (perimesonephric and iliac) [15].

Centrifugal sprouting of lymphatic vessels from the lymphatic sac then generates the peripheral lymphatic vasculature (Figure 1C), with the fusion of the separate lymphatic capillary networks (Figure 1D), which finally spread into the thorax, upper limbs, neck, and head [18], followed by the remodeling and maturation of the primitive lymphatic capillary plexus (Figure 1E). The abdominal viscera and diaphragm are reached by the capillary plexus and lymphatic vessels spread from the retroperitoneal lymph sac (perimesonephric region) [15], while the last (posterior) lymphatic sac develops from the iliac veins spreading capillary networks and the lymphatic vessels to the abdominal wall, the pelvic region, and the lower limbs. These lymphatic vessels then join the chyle cistern and lose their connections with the adjacent veins. All sacs are invaded by mesenchymal cells and are converted into groups of lymph nodes, except for the anterior part of the sac, from which the chyle cistern develops [2]. Moreover, numerous intussusceptive pillars are formed by lymph node sinuses emerging from the lymphatic sac, creating a meshwork of intra-sinusal processes between the opposite walls of the sinuses [19]. This particular lymphatic vascular formation occurs also in an experimental therapeutic model of lymphedema reduction, where lymphatic vessels are created starting from adipose-derived stem cells (ADSCs) [20].

The endothelial cell subpopulation that gives rise to the mammalian lymphatic system (lymphatic endothelial cells, LEC) expresses the homeobox Prox 1 gene (Figure 1B). In fact, in Prox 1-null embryos, this budding stops around the embryonic stage in E11.5 in mice, resulting in embryos without lymphatic vascularization. Unlike endothelial cells that proliferate in wild-type embryos, those of Prox 1-null embryos do not co-express typical lymphatic markers such as vascular endothelial growth factor receptor-3 (VEGFR-3) or lymphatic vessel endothelial hyaluronan receptor 1 (LYVE-1). Moreover, mutant cells appear to have a vascular phenotype, determined by the expression of laminin and CD34. This finding suggests that the activity of Prox-1 is required both for the maintenance of the budding of venous endothelial cells and for differentiation towards the lymphatic phenotype [18]. LYVE-1, a homolog of CD44, is the major cell surface receptor for hyaluronic acid (HA) in LEC [21]. HA is a large non-sulfur glycosaminoglycan found in the extracellular matrix. Low-molecular-weight hyaluronan (LMW-HA), formed by HA-degraded fragments, is an essential regulator of angiogenesis [21]. Neutralizing anti-LYVE-1 antibodies block the binding interaction between LMW-HA and LYVE-1 and inhibit LMW-HA-induced proliferation, migration, tubing, and signal transduction in LEC, suggesting that LMW-HA plays a critical role in lymphangiogenesis [21]. The lymphangiogenesis process is mainly regulated by the vascular endothelial growth factor (VEGF) family and their receptors and ligands. In mammals, the VEGF family comprises five members: VEGF-A, placenta growth factor (PGF), VEGF-B, VEGF-C, and VEGF-D and three main subtypes of VEGF receptors (VEGFR) numbered 1, 2, and 3, which may be membrane-bound (mbVEGFR) or soluble (sVEGFR), depending on alternative splicing [22]. VEGF-C and D are more involved in lymphangiogenesis, while VEGF-A is more involved in angiogenesis [15,23,24] (Figure 1).

The VEGF-A and VEGF-B isoforms are produced through alternative splicing, while the different forms of VEGF-C and VEGF-D are produced by the proteolysis of precursor proteins, activated by intracellular proprotein convertases [15,23,24]. Secreted disulfide-linked VEGF-C subunits selectively bind VEGFR-3, but their extracellular proteolysis by plasmin and other proteases generates disulfide-unlinked homodimeric proteins with high affinity for both VEGFR-2 and VEGFR-3 [15,24]. Both VEGF-C and VEGF-D induce the proliferation, migration, and survival of endothelial cells [25]. VEGF-C is expressed mainly in lymphatic vessel regions during development [24,26]. When VEGF-C has been inactivated in Xenopus tadpoles, zebrafish, and mice, LECs differentiate in the embryonic veins but fail to migrate to form the primary lymph sacs [26,27,28]. Furthermore, in mouse embryos, VEGF-C homozygous deletion leads to the complete absence of a lymphatic vascular system, while severe lymphatic hypoplasia has been observed in VEGF-C heterozygous mice, indicating that, on the contrary, the development of the lymphatic vasculature in mice is not affected by VEGF-D deletion [26,29]. Contemporary VEGF-C and VEGF-D deletion fails to induce the early embryonic lethality observed in VEGFR-3-null mice [30]; therefore, it is possible to hypothesize the presence of other VEGFR-3 activator factors. However, VEGF-C/VEGFR-3 signaling plays a key role in development of the lymphatic vascular tree after lymph sac formation. In fact, it has been observed that, in mouse embryos from E14.5 onward, overexpression of soluble VEGFR-3-immunoglobulin G, Fc-domain fusion protein, a real VEGF-C/D Trap, results in severe lymphatic vessel hypoplasia [31]. VEGF-C/D Trap or VEGFR-3-blocking monoclonal antibodies are also able to induce the temporary regression of lymphatic vessels developed in the first two postnatal weeks, as they regrow at 4 weeks of age despite sustained VEGFR-3 inhibition [32,33]. Moreover, VEGF-C/VEGFR-3 signaling seems not required for the maintenance of the lymphatic vasculature in adulthood [34].

It has been also described that both VEGF-C and VEGF-D bind to the axon guidance receptor neuropilin-2 (NP-2) that is expressed in veins and lymphatic vessels and internalized with VEGFR-3 upon stimulation with the ligand [35]. NP-2 and VEGFR-3 increase the affinity of LECs for VEGF-C/D; NP-2-blocking antibodies arrest the VEGF-C-stimulated lymph sprout elongation without affecting further lymphatic vessel growth [36], and, finally, NP2 mutant mice showed lymphatic capillary hypoplasia [37]. Overexpression of VEGF-C or VEGF-D or their VEGFR-3-specific forms in adult tissues stimulates lymphangiogenesis. Importantly, damaged collecting lymphatic vessels can also be regenerated via lymphatic capillaries undergoing an intrinsic maturation program in response [24].

The receptor tyrosine kinase vascular endothelial growth factor receptor-3 (VEGFR-3) was one of the first lymphatic endothelial markers to be discovered [15,24] (Figure 1A–C). VEGFR-3 is activated by VEGF-C and VEGF-D, both members of the VEGF growth factor family [23,24]. VEGFR-3 forms heterodimers with VEGFR-2 upon binding of the proteolytically processed forms of VEGF-C and VEGF-D, which leads to unique combinatorial signals from the intracellular domains of the two receptors [38]. During the early stages of development, VEGFR-3 is present in all endothelia and VEGFR-3-null mice die around E10.5 due to impaired development of the cardiovascular system [15,24]. High levels of VEGFR-3 are expressed by lymphatic lineage-committed endothelial cells, but VEGFR-3 expression becomes exclusively restricted to the LEC as the development of the lymphatic vasculature begins. An exception is represented by fenestrated blood vessels present in endocrine organs [25].

Vascular endothelial growth factor-A (VEGF-A) expression has been reported to be upregulated in damaged skin tissue, where it promotes angiogenesis. In transgenic mice overexpressing VEGF-A in the epidermis, VEGF-A-dependent lymphangiogenesis was observed at the site of wound healing. Studies in cultured lymphatic endothelial cells also revealed that VEGF-A induces the expression of integrins α1 and α2, which in turn promote hapotactic migration to type I collagen [39]. Therefore, it is likely that VEGF-A/induces lymphatic vessel formation via VEGFR-2 and distinct dynamics of wound-associated angiogenesis and lymphangiogenesis through lineage-specific differences in integrin receptor expression [39].

## 3. Lymphatic System in Wound Healing

The healing of a skin wound is a complex process composed of several overlapping phases (Figure 2), which, if well orchestrated by the body, lead to the restoration of the skin and vascularization to a healthy and functional condition (Figure 2C). This delicate sequence of events may fail to occur completely in diabetic or elderly patients [40,41,42], where healing does not proceed through all stages, stopping at the stage of inflammation (Figure 2B) and resulting in a chronic wound [43,44] (Figure 2D). Wound healing involves also lymphangiogenesis, which has acquired an increasingly clear role in recent years [45,46,47,48].

Over the past 40 years, the role of the lymphangiogenesis process in wound healing has been little investigated, while angiogenesis during this process has received enormous attention. This oversight was mainly due to the lack of specific markers for the identification of lymphatic vessels; a renewed interest in lymphangiogenesis was observed when VEGFR-3 was characterized, and it became a lymphatic marker that allowed the quantifiable and precise observation of lymphatic dynamics [3,49]. Although VEGFR-3 is also expressed to some extent in newly formed blood capillaries and the fenestrated endothelium, it is still one of the best markers for all lymphatic endothelial cells.

This research oriented towards the lymphatic system has allowed important discoveries, such as that relating to the presence of lymphatic vessels in some areas of the central nervous system [50]. A further impetus in the study of lymphatic regeneration was provided by the hypothesis of a central role of lymphangiogenesis in promoting the healing of chronic wounds [3,51,52,53], since it has been observed that delayed or failed lymphatic regeneration (as in diabetic patients) is one of the main causes of impaired healing [42,54,55,56] (Figure 2D and Figure 3).

Lymphatic vessels are both pressure regulators in tissues and vectors of the immune response, as an exit route for T lymphocytes, macrophages, and Langerhans cells, particularly in the context of wound healing [3,51,52,53,57,58]. Lymphangiogenesis can be abundant in association with pathological processes such as edema; in fact, it typically occurs in sites of inflammation and is induced by factors produced by macrophages and granulocytes. The process contributes to the reduction of tissue edema and the activation of immune responses by facilitating the drainage of fluids and the transport of dendritic cells and macrophages.

Indeed, during the wound healing inflammatory phase, leukocytes and macrophages secrete VEGF-C/D, which bind to VEGFR-3 and VEGFR-2 expressed on the surface of the LEC to promote lymphangiogenesis (Figure 2). This binding signals Akt phosphorylation and MAPK-activated p42/p44 protein kinase activation, resulting in LEC survival and migration. Increased VEGF signaling regulates lymphatic growth and survival, while decreased VEGF signaling causes lymphatic regression [59,60] (Figure 3).

An inflammatory phase is required for normal wound healing, but this process is abnormal in diabetes. In fact, in diabetic wounds, impaired inflammatory cell function, decreased secretion of cytokines/growth factors, and a prolonged inflammatory phase have been observed [54,61]. In fact, a reduced macrophage number has been observed in diabetic mice, and these cells have been shown to secrete reduced levels of inflammatory cytokines such as IL-1β and tumor necrosis factor-α [55,62], which stimulate the release of VEGF-C/D [56]. Therefore, an inadequate macrophage number could be one factor contributing to the reduced lymphatic structures in the granulation tissue of wounds and to impaired wound healing (Figure 3).

Finally, two main hypotheses have been advanced to describe the lymphangiogenesis process in wound healing from a morphological point of view:(1)a “self-organization” process in which single LECs migrate into the wound following the interstitial fluid flow and begin to self-organize into capillary structures only after reaching a certain density of threshold [63];(2)a “germination” process, in which adult lymphangiogenesis occurs primarily from pre-existing capillary tips; the capillary tips are attached to the ends of the vessels, and, therefore, contrary to the LECs, they are not subject to the advection of the interstitial fluid [60,64].

For both cases, immediately after the wound, the transformation factor β (TGF-β) is activated and chemotactically attracts the macrophages to the wound, which in turn secrete VEGF, which induces capillary regeneration by acting on both the LECs (in the case of self-organization) and on the tips of the capillaries (in the case of germination).

Lymphangiogenesis in shallow wounds is dominated by growth/remodeling and it occurs symmetrically on both sides of the wound. In this case, there is little or no difference between the two hypotheses related to wound healing lymphangiogenesis dynamics.

A definitive answer to the question of whether the self-organization or germination hypothesis better describes lymphangiogenesis requires more detailed spatiotemporal measurements of capillary density and chemical concentrations. Moreover, the type of wound (superficial or deep) strongly influences the speed and shape of the regeneration process: deeper wounds take longer to heal, and lymphangiogenesis will occur more markedly in the direction of the lymphatic flow [46].

## 4. Lymphangiogenesis Induction to Accelerate Wound Healing

As the role of lymphangiogenesis in wound healing becomes more and more evident, the question of its induction in the case of chronic wounds in order to increase healing may also arise.

Despite the importance of lymphangiogenesis as a therapeutic target, there are still few experimental models to trace and study this process in vivo. An example, however, are the different lines of transgenic mice used for the fluorescent visualization of the recently reported lymphatic vessels. All these lines are based on BAC transgenic constructs targeted to the gene to express GFP, mOrange, or Tomato, fluorescent proteins under the transcriptional control of Prox-1, using VEGFR-3 as a lymphatic marker. The final result are knock-IN mice for the in vivo imaging of lymphatic vessels and lymphangiogenesis based on the double fluorescence and luminescence emanating from the signal/reporter allele (EGFPLuc: a coding sequence preceded by an IRE element then introduced at the level of the 3’-UTR region of the mouse VEFGR3 locus by gene targeting) [45].

A possible therapeutic innovation could be the use of stem/progenitor cells in the regeneration of lymphatic vessels [65]. This type of approach has been proposed for the treatment of lymphedema, a condition caused by the deficiency of the lymphatic system. Even if it remains undetermined whether stem/progenitor cells support a complex regenerative response across the entire anatomical spectrum of the system, multipotent adult progenitor cell (MAPC) transplantation supports the growth of blood vessels and lymphatic capillaries in wounds by stimulating the regeneration of capillaries and pre-collector vessels. Therefore, MAPC transplantation represents a promising remedy for restoring the lymphatic system at different anatomical levels.

The application of pharmacological concentrations of purified polypeptide growth factors, cytokines, and matrix molecules has resulted in the acceleration of normal repair in a wide variety of skin wound models. It would be interesting to consider the use of drugs that can modulate lymphangiogenesis. At the moment, there are both drugs capable of inhibiting lymphangiogenesis, mainly used for the treatment of neoplasms, and drugs capable of increasing this phenomenon on the market. Below is a short list of these drugs, divided into inhibitors and inducers of lymphangiogenesis. Thus far, different drugs have shown effects.

### 4.1. Simvastatin

Simvastatin is an inhibitor of 3-hydroxy-3-methyl-glutaryl-coenzyme A (HMG-CoA) reductase, belonging to the statin class of drugs. Simvastatin is used for lipid lowering in cardiovascular diseases. Recently, its therapeutic effects, beyond plasma cholesterol lowering, have been found in terms of fracture healing [66] and anti-inflammatory activity on skin [67]. In particular, it has been demonstrated that the topical application of simvastatin promotes wound healing in diabetic mice by augmenting angiogenesis and lymphangiogenesis [42]. Studies have evidenced that simvastatin is able to exert multiple effects on lymphatic endothelial cells, both on metabolic pathways and on stimulating factors. In fact, simvastatin stimulates the AKT/PI3K/mTOR pathway, which represents an important pathway for several biological functions of LECs. Furthermore, simvastatin has been observed to stimulate macrophages to secrete VEGF-C, which represents a fundamental growth factor in inducing lymphatic proliferation. Furthermore, simvastatin is able to induce the transformation of M1-phenotype pro-inflammatory macrophages into M2-phenotype anti-inflammatory/tissue reparative macrophages, and reduce oxidative stress-induced apoptosis [68]. In conclusion, the topical application of simvastatin may have significant therapeutic potential for improving wound healing in patients with impaired microcirculation, such as diabetics.

### 4.2. COMP-Angiopoietin 1

Angiopoietin-1 (Ang1) is a specific growth factor that has the function of generating stable and functional vascularization through the Tie2 and Tie1 receptors. Delayed skin wound healing is a serious complication accompanied by impaired cutaneous blood flow, hypoxia, accelerated inflammation, edema, and endothelial–neural dysfunction [69]. Therefore, restoring the structural and functional microvasculature via the supplementary delivery of Ang1 could be beneficial to increase the restoration of vascular regeneration and functionalization. Cho and colleagues [53] investigated the efficacy of oligomeric cartilage matrix (COMP)-Ang1, as a soluble, stable, and potent form of Ang1, on promoting the healing of skin wounds in diabetic mice. It was observed that mice treated with the COMP-Ang1 protein showed accelerated wound closure and epidermal and dermal regeneration, following the potent stimulation of angiogenesis. Furthermore, the authors have observed that angiogenesis is accompanied by also lymphangiogenesis, which characterizes strongly the region in which the wound healing takes place. Interestingly, the research indicates that COMP-Ang1 can promote wound healing in diabetes through enhanced angiogenesis and blood flow and lymphangiogenesis, which became a fundamental characteristic of wound healing [53].

### 4.3. Retinoic Acids

Retinoic acids (RAs) are composed of biologically active metabolites of vitamin A and are involved in a broad range of biological processes in vertebrate development by regulating genes important for cell proliferation, differentiation, apoptosis, and metabolism [70,71]. RAs have been shown to arrest cell cycle progression and cell proliferation in several cell types. However, the proangiogenic effect of RAs has also been documented, and RAs have been observed to stimulate the transcription and translation of vascular endothelial growth factor (VEGF) in nonendothelial cells. Recently, all-trans RA has been shown to be important in the early steps of lymphatic development during embryogenesis [72]. Furthermore, experiments have indicated that RAs are able also to potently induce post-developmental lymphatic regeneration. In particular, a specific RA metabolite, -cis retinoic acid (9-cisRA), is able to induce lymphangiogenesis, but not angiogenesis. The 9-cisRA seems to represent a specific inductor of the proliferation and differentiation of LECs through both genomic and nongenomic actions. The transcription-dependent genomic action of 9-cisRA consists of the regulation of PROX1 gene expression. The nongenomic action consists of the downregulation and phosphorylation of p27^Kip1^. These findings make RAs a valid tool to assist in wound healing, whereas, in particular, 9-cisRA becomes a promising therapeutic agent for the specific treatment of patients with lymphedema [49].

### 4.4. CCBE1

Vascular endothelial growth factor C (VEGF-C) is a key factor in promoting lymphatic endothelial cells’ proliferation and migration by its receptor VEGFR-3. VEGF-C is secreted as a monomeric 58 kDa precursor; it is first proteolytically processed to a 43 kDa polypeptide, and then C-terminally processed to the 29/31 kDa pro-VEGF-C form and finally fully processed to the 21 kDa mature form. The several forms produced have different degrees of receptor-binding capacity for VEGFR-3 and VEGFR-2, with the last proteolytic form having the highest activity towards VEGFR-3 [73,74].

Recent studies have indicated collagen and calcium-binding EGF domain-1 (CCBE1) as the key factors involved in promoting VEGF-C proteolysis by the A disintegrin and metalloproteinase with thrombospondin motifs 3 (ADAMTS3) [74,75]. In particular, Song and colleagues [48] have evidenced, in a colon cancer experimental model, that CCBE1 enhances VEGF-C and facilitates tube formation and the migration of LECs, promoting tumor lymphangiogenesis and consequently the lymphatic metastasis of colon cancer cells. Furthermore, the same authors have evidenced that transforming growth factor beta (TGF-β) downregulates the transcription and lymphangiogenic function of CCBE1 through the direct binding of SMAD proteins (they modulate the activity of transforming growth factor beta ligands) at the gene locus by CCBE1. Even if the study of Song and colleagues is focused on colon cancer tumor, and the role of lymphangiogenesis in promoting tumor metastasis via lymphatic vessels, nevertheless, more generally, these findings demonstrate the role of CCBE1 in promoting lymphangiogenesis and the role of TGF-β signaling as a regulatory factor [47].

### 4.5. circEHBP1

VEGF-C and VEGF-D play a key role in embryonal lymphangiogenesis. VEGF-C accompanies the first step in the migration and formation of tubular elements that characterizes the lymphatic plexus, whereas VEGF-D intervenes in the last step of lymphatic plexus maturation, perhaps performing and assisting the functional aspect of lymph flow in vessels. Zhu and colleagues [48] have revealed that the formation of new lymphatic vessels in bladder cancer (BCa) is exclusively VEGF-D-dependent. The authors have observed that circEHBP1, a circular RNA, is upregulated in bladder cancer and positively correlated with lymphatic metastasis and poor prognosis. Circular RNAs (circRNAs) consist of a class of single-stranded molecules with development-specific expression patterns; they are characterized by a covalently closed structure and generated from the back-splicing of pre-mRNA transcripts. In bladder cancer, circEHBP1 sponges miR-130-3p, which has a suppressive effect on lymphangiogenesis in BCa. miR-130-3p regulates the expression of the TGF-β receptor (TGFβR1), inhibiting its expression. TGFβR1 is a phosphorylase of the TGF-β/SMAD3 signaling pathway; this pathway is essential in inducing specifically the expression of VEGF-D and promoting lymphangiogenesis [48].

In summary, circEHBP1 induces in BCa lymphangiogenesis, promoting the expression of VEGF-D, by sponging miR-130-3p, giving free expression to TGFβR1.

## 5. Future Perspectives

The role of lymphangiogenesis in wound healing is a field of investigation that has received considerable attention in recent years thanks to two important factors: the use of increasingly reliable markers to identify the various subpopulations of lymphatic endothelial cells and the use of an increasing number of drugs for the modulation of lymphangiogenesis.

In fact, after the discovery of the first true lymphatic markers, such as LYVE-1, other new, reliable markers have been discovered in recent years, such as cir-cEHBP1, which is presented as a potential biomarker and promising therapeutic target for lymphatic metastases in bladder cancer and CCBE1, whose expression may be taken as an accurate prognostic marker for colorectal cancer.

We can therefore state that we are only at the beginning of an important phase in the study of the role of lymphangiogenesis, both in normal and in pathological processes. The induction of lymphangiogenesis by pharmacological and/or cellular stimulation is a current target to fight secondary lymphedema, a common complication of lymph node dissection [76], and it could prove also to be important in the treatment of chronic wounds caused by metabolic diseases such as diabetes, opening up a new therapeutic area that has not been explored so far. In light of the recent literature analyzed in this paper, it is possible to state that altered lymphangiogenesis participates, along with other phenomena, in slowing or inhibiting the normal wound healing process. However, there is the possibility of positively modulating this phenomenon through the topical administration of experimental drugs at the wound site, avoiding collateral systemic effects, and it is reasonable to assume that, in the next few years, it will be possible to witness a significant increase in the number of clinical studies that will consider the use of inducers with regard to lymphangiogenesis as promoters of wound healing.

## Figures and Tables

**Figure 1 biomedicines-11-00656-f001:**
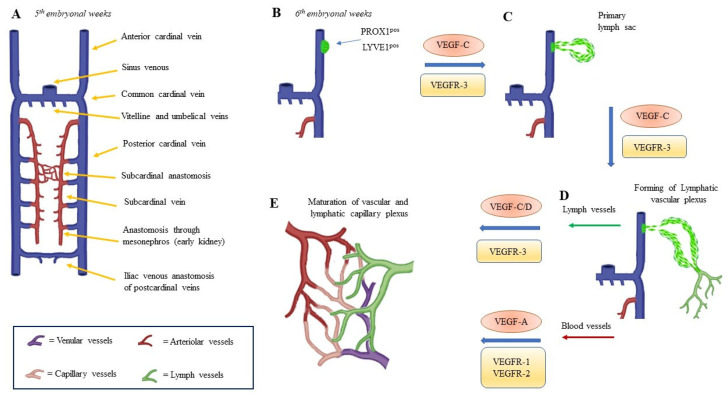
Embryonal development of lymphatic vessels. (**A**) Structural scheme of embryonal vein vascular organization at 5th week of development in humans. Following formation of anterior cardinal vein, lymphatic vessels develop from the vein at the 6th week (**B**) and, through the formation of a primitive vascular sac (**C**), evolve to an independent vascular plexus, via the expression of the selective action of growth factors and related receptors (**D**). Finally, angiogenic and lymphoangiogenic factors coordinated action allows the creation of vascular and lymphatic capillary plexus (**E**).

**Figure 2 biomedicines-11-00656-f002:**
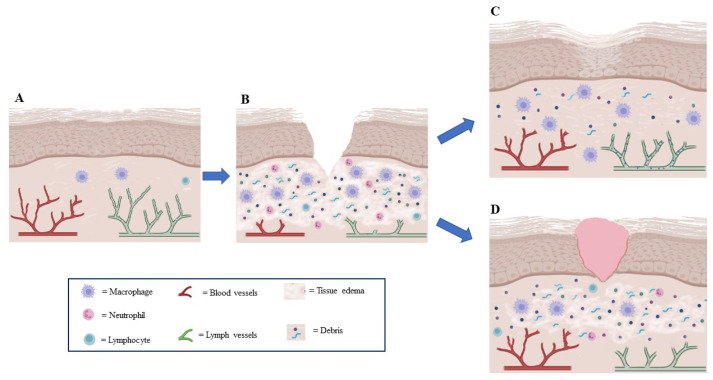
Lymphangiogenesis role in wound healing. (**A**) In normal tissue, lymphatic vessels drain fluid extravasated from blood vessels and containing macromolecules and white blood cells (macrophages and lymphocytes) so that interstitial pressure does not increase. (**B**) In wounds, lymphatic and blood vessels are damaged and lymphatic drainage is insufficient, thus creating edema; an inflammatory phase will be created, which is necessary for the triggering of the following phases of wound healing. (**C**) During normal wound healing, lymphangiogenesis increases concurrently with angiogenesis to drain extravasated fluid from leaking new vessels, thus preventing excessive lymphedema and reducing the presence of inflammatory cells. (**D**) When lymphatic vessels are hypoplastic, lymph drainage is insufficient, resulting in lymphedema and increased interstitial tissue pressure, chronic inflammation, poor vascularization, and delayed or absent healing.

**Figure 3 biomedicines-11-00656-f003:**
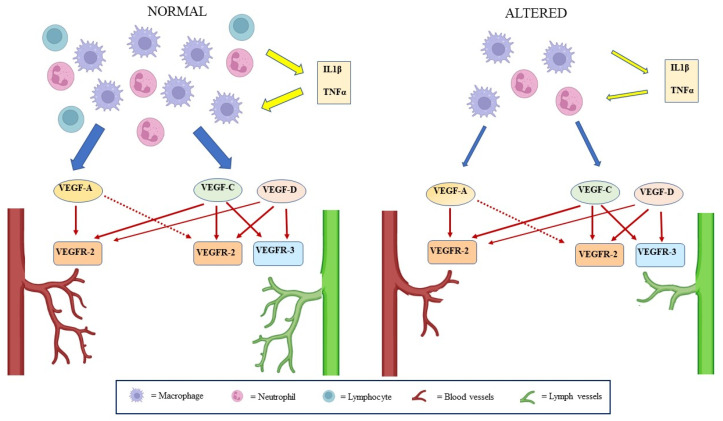
Lymphangiogenesis mechanisms in wound healing. Concomitant angiogenesis and lymphangiogenesis occur in response to tissue damage. During the transient inflammatory phase of wound healing, in the presence of pro-inflammatory cytokines (IL-1β, TNFα), macrophages, lymphocytes, and monocyte-derived granulocytes secrete several members of the VEGF family with direct or indirect lymphangiogenic and direct angiogenic effects. In the case of chronic inflammation, both angiogenic and lymphangiogenic stimulation are compromised with local edema, which delays or hinders the resolution of the inflammation.

## Data Availability

The data presented in this study are available from the corresponding author upon reasonable request.

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
