# Peer review of "Breaking a Vicious Circle: Lymphangiogenesis as a New Therapeutic Target in Wound Healing"

_biomedicines, 2023, doi:10.3390/biomedicines11030656_

Round 1
Reviewer 1 Report
The review focused on the lymphangiogenesis as a new therapeutic target in wound healing. These findings are very impressive, however, some issues required to be addressed.
The author must provide the paper of the lymphangiogenesis as a new therapeutic target in wound healing in vivo.
Author Response
The author must provide the paper of the lymphangiogenesis as a new therapeutic target in wound healing in vivo.
Lymphoangiogenesis has been considered as a therapeutic target in vivo wound healing in the following articles cited in the first submitted version
n.39 Hong, Y et al. VEGF-A promotes tissue repair-associated lymphatic vessel formation via VEGFR-2 and the alpha1beta1 and alpha2beta1 integrins. FASEB J. 2004, 18, 1111-3.
n.42 Asai, J et al., Topical simvastatin accelerates wound healing in diabetes by enhancing angiogenesis and lymphangiogenesis. Am. J. Pathol. 2012, 181, 2217-2224.
n.49 Cho, C et al., COMP-angiopoietin-1 promotes wound healing through enhanced angiogenesis, lymphangiogenesis, and blood flow in a diabetic mouse model. Proc. Natl. Acad. Sci. U.S.A. 2006, 103, 4946-51.
n.54 Saaristo, A et al., Vascular endothelial growth factor-C accelerates diabetic wound healing. Am. J. Pathol. 2006, 169, 1080-7.
n.63 Rutkowski, J.M. et al., Characterization of lymphangiogenesis in a model of adult skin
regeneration. Am. J. Physiol. Heart. Circ. Physiol. 2006, 291, H1402-10.
n.65 Beerens, M.et al., Multipotent Adult Progenitor Cells Support Lymphatic Regeneration at Multiple Anatomical Levels during Wound Healing and Lymphedema. Sci. Rep. 2018, 8, Article number: 3852.
n.67 Adami, M., Simvastatin ointment, a new treatment for skin inflammatory conditions. J. Dermatol. Sci. 2012, 66,127-135.
Reviewer 2 Report
The authors described "Breaking a vicious circle: lymphangiogenesis as a new therapeutic target in wound healing" in a review style. This topic should be attractive for potential readers and emerging in wound repair regeneration.
I recommend they should add the topic regarding "intussusceptive lymphangiogenesis". In adult tissue, intussusceptive lymphangiogenesis is occuring (Ogino R et al. IJMS. 2020, Díaz-Flores L et al. Histol Histopathol. 2020). So, this topic and figures should be added.
In addition, recent pharmacotherapy, and cell therapy for lymphatic disorders should be added in future perspectives because this manuscript is "Review".
Author Response
I recommend they should add the topic regarding “intussusceptive lymphangiogenesis”. In adult tissue, intussusceptive lymphangiogenesis is occuring (Ogino R et al. IJMS. 2020, Díaz-Flores L et al. Histol Histopathol. 2020). So, this topic and figures should be added.
Intussusceptive lymphangiogenesis concept has been added along with the two suggested refernces to the “Embryology and development of lymphatic vasculature “ paragraph (pg.3, line 94-100). Authors decided not to add more particulars to the figure 1 because of its complexity.
In addition, recent pharmacotherapy, and cell therapy for lymphatic disorders should be added in future perspectives because this manuscript is a “Review”.
In the chapter “Future perspectives” (pg.14, line 468-469) a new sentence and related bibliographic reference has been added (Ogino R et al., Anti-Inflammatory Pharmacotherapy and Cell-Based Therapy for Lymphedema. Int J Mol Sci. 2022 Jul 9;23(14):7614. doi: 10.3390/ijms23147614. PMID: 35886961; PMCID: PMC9322118.)
Round 2
Reviewer 1 Report
This paper is accepted for publication
Reviewer 2 Report
The authors revised the manuscript. And this should be attractive. Thank you for this opportunity.